# Circulating Tumour Cell Associated MicroRNA Profiles Change during Chemoradiation and Are Predictive of Response in Locally Advanced Rectal Cancer

**DOI:** 10.3390/cancers15164184

**Published:** 2023-08-20

**Authors:** Stephanie H. Lim, Wei Chua, Weng Ng, Emilia Ip, Tania M. Marques, Nham T. Tran, Margarida Gama-Carvalho, Ray Asghari, Christopher Henderson, Yafeng Ma, Paul de Souza, Kevin J. Spring

**Affiliations:** 1Medical Oncology Group, Ingham Institute for Applied Medical Research, Liverpool, NSW 2170, Australia; wei.chua@health.nsw.gov.au (W.C.); weng.ng@health.nsw.gov.au (W.N.); emilia.ip@health.nsw.gov.au (E.I.); yafeng.ma@unsw.edu.au (Y.M.); p.desouza@westernsydney.edu.au (P.d.S.); 2Department of Medical Oncology, Macarthur Cancer Therapy Centre, Campbelltown, NSW 2560, Australia; 3Liverpool Clinical School, Western Sydney University, Liverpool, NSW 2170, Australia; 4Department of Medical Oncology, Liverpool Hospital, Liverpool, NSW 2170, Australia; 5BioISI—Biosystems & Integrative Sciences Institute, Faculty of Sciences, University of Lisbon, 1749-016 Lisbon, Portugal; tania.marques.mt@gmail.com (T.M.M.); mhcarvalho@ciencias.ulisboa.pt (M.G.-C.); 6School Biomedical Engineering, Faculty of Engineering and IT, University of Technology Sydney, Ultimo, NSW 2007, Australia; nham.tran@uts.edu.au; 7Department of Medical Oncology, Bankstown Hospital, Bankstown, NSW 2200, Australia; ray.asghari@health.nsw.gov.au; 8NSW Health Pathology, Liverpool Hospital, Liverpool, NSW 2170, Australia; christopher.henderson@health.nsw.gov.au; 9School of Medicine, University of Wollongong, Wollongong, NSW 2522, Australia; 10South West Sydney Clinical School, University of New South Wales, Liverpool, NSW 2170, Australia

**Keywords:** locally advanced rectal cancer, chemoradiotherapy, microRNA, circulating tumour cells, lymphocytes, predictive biomarkers

## Abstract

**Simple Summary:**

A standard treatment for locally advanced rectal cancer consists of trimodality therapy of neoadjuvant radiation +/− chemotherapy, surgery, and adjuvant chemotherapy. There is a need for biomarkers to predict treatment response and outcome with the aim of personalizing patient treatment. We analysed a class of promising biomarkers in the blood, namely circulating tumour cells and the expression of a panel of selected small RNAs (microRNAs) which are known to regulate genes and how cancers behave. We identified these cells in a majority of patients assessed in this study, and also determined microRNA expression changes during the course of treatment. Some of these changes were specifically associated with responses to treatment and could be potentially used to predict how patients respond to treatment. They may also be potential targets for drug development.

**Abstract:**

Locally advanced rectal cancer (LARC) has traditionally been treated with trimodality therapy consisting of neoadjuvant radiation +/− chemotherapy, surgery, and adjuvant chemotherapy. There is currently a clinical need for biomarkers to predict treatment response and outcomes, especially during neoadjuvant therapy. Liquid biopsies in the form of circulating tumour cells (CTCs) and circulating nucleic acids in particular microRNAs (miRNA) are novel, the latter also being highly stable and clinically relevant regulators of disease. We studied a prospective cohort of 52 patients with LARC, and obtained samples at baseline, during treatment, and post-treatment. We enumerated CTCs during chemoradiation at these three time-points, using the Isoflux^TM^ (Fluxion Biosciences Inc., Alameda, CA, USA) CTC Isolation and detection platform. We then subjected the isolated CTCs to miRNA expression analyses, using a panel of 106 miRNA candidates. We identified CTCs in 73% of patients at baseline; numbers fell and miRNA expression profiles also changed during treatment. Between baseline and during treatment (week 3) time-points, three microRNAs (hsa-miR-95, hsa-miR-10a, and hsa-miR-16-1*) were highly differentially expressed. Importantly, hsa-miR-19b-3p and hsa-miR-483-5p were found to correlate with good response to treatment. The latter (hsa-miR-483-5p) was also found to be differentially expressed between good responders and poor responders. These miRNAs represent potential predictive biomarkers, and thus a potential miRNA-based treatment strategy. In this study, we demonstrate that CTCs are present and can be isolated in the non-metastatic early-stage cancer setting, and their associated miRNA profiles can potentially be utilized to predict treatment response.

## 1. Introduction

A standard treatment for locally advanced rectal cancer (LARC) is trimodality therapy, consisting of neoadjuvant radiation +/− chemotherapy, surgery, and adjuvant chemotherapy. Despite trimodality therapy, around a third of patients still develop recurrent disease [1]. Currently, there are no biomarkers in routine clinical practice for patients with LARC to personalise treatment. Predictive markers are of particular relevance in the neoadjuvant chemoradiation phase, as early assessment may mean patients who experience a complete response may be spared unnecessary surgery or additional chemotherapy. Conversely, early non-response may indicate the need to escalate therapy. This is particularly relevant given the recent move towards total neoadjuvant therapy (TNT) and the watch and wait approach for complete responders [2,3]. In current practice, response and prognosis are assessed histologically at time of surgery by pathological stage (ypT and ypN) and tumour regression grade (TRG) as it has been demonstrated that pathological complete response is associated with superior survival [4].

Liquid biopsies in the form of circulating tumour cells (CTCs) or circulating nucleic acids (ctDNA and ctRNA) offer the promise of earlier detection of resistance to treatment whilst being relatively non-invasive. Serial liquid biopsies may also allow clinicians to monitor tumour biology and adapt treatments accordingly. There are various methodologies used in isolating CTCs [5]. One of the most widely used approaches for CTC isolation and enumeration involves immunomagnetic based separation, utilising EpCAM-antibody-coupled magnetic beads to isolate epithelial cells from peripheral venous blood. CTCs are routinely identified as pan-cytokeratin (CK 8, 18, 19) and DAPI (4,6 diamidino-2-phenylindole) positive, and CD45 negative (leukocyte common antigen). CTCs derived using this approach also result in co-isolation of associated leucocytes. 

Circulating miRNAs are increasingly becoming a focus of research, as they are highly stable non-coding RNAs thought to regulate up to 30% of the human genome and are potentially key regulators in rectal cancer [6]. The rectal cancer microRNAome was defined by Gaedcke et al. after extensive microarray-based miRNA profiling of LARC pre-treatment biopsy samples [7].

The aims of this study were to prospectively assess CTC change and to determine CTC/lymphocyte miRNA expression during the course of neoadjuvant chemoradiation for LARC. Finally, we correlated CTC/lymphocyte miRNA expression levels and change with tumour treatment response. To the best of our knowledge, our study is the first to describe miRNA profiles associated with CTCs isolated immunomagnetically during neoadjuvant chemoradiation in LARC.

## 2. Materials and Methods

### 2.1. Patient Selection and Sample Time-Points

Patients were prospectively recruited from three centres in the South Western Sydney Local Health District, Australia. The inclusion criteria were: age > 18; tissue confirmation of LARC (rectal adenocarcinoma stage T3/T4 or node positive disease and no evidence of metastatic disease on computed tomography imaging of chest/abdomen/pelvis); and planned treatment with neoadjuvant chemoradiation. Neoadjuvant chemoradiation consisted of 5-fluorouracil (5-FU) infusion or oral capecitabine concurrent with radiotherapy for 5–6 weeks. This was followed by surgery approximately 6–12 weeks after completion of chemoradiation. Patients were followed up in clinic every 3 months. Recurrence was defined as the appearance of local or distant disease. Ethical approval was obtained from the Sydney South West Area Health Service Ethics Review Committee (reference number HREC/13/LPPL/158, date of approval 2 September 2013). CTCs were collected at baseline (pre-treatment), during treatment (at week 3), and post-treatment (within 1 week pre-surgery). Two blood samples were collected using K2EDTA BCTs, the first for enumeration and the second for miRNA analyses. 

### 2.2. TRG Scoring

Tumour Regression Grade (TRG) and pTNM staging (American Joint Committee on Cancer AJCC 7th edition/modification of Ryan et al.) were scored by two specialist pathologists on the surgical specimens [8,9]. All discrepancies between pathologists were resolved through consensus. The system used for TRG scoring was the AJCC 4-point scale adapted from Ryan et al. with 0 being complete tumour response with no viable tumour cells, 1 being small groups of cancer cells, 2 being residual cancer outgrown by fibrosis, and 3 being poor response with extensive residual tumour. 

### 2.3. CTC Sample Processing

A 9 mL blood sample was drawn in a K2EDTA tube and processed within 24 h using the Isoflux^TM^) system, as per the manufacturer’s instructions. For enumeration, immunostaining was performed with anti-CD45 for negative selection of lymphocytes, pan-cytokeratin (CK) for positive selection of epithelial cells and Hoechst for DAPI/nuclear staining. For miRNA analyses, the isolated cell pellet was placed in a cryovial, snap frozen on dry ice, and stored in liquid nitrogen until RNA extraction.

### 2.4. miRNA Candidate Selection

Three approaches were combined to select candidate miRNAs for OpenArray^®^ profiling: (i) A review of the literature involving miRNAs previously associated with LARC treatment, with a focus on studies that described change in expression during treatment and association with response; (ii) a review of miRNAs previously shown to be associated with colorectal cancer carcinogenesis and radiosensitivity, and (iii) additional miRNAs identified in a pilot study of rectal cancer patient CTC samples, SW480 cell line, and healthy patient samples using an OpenArray^®^ panel with 758 potential target miRNAs. The candidates from this run were compared against those derived from the literature. 

### 2.5. RNA Extraction and miRNA OpenArray Profiling

RNA extraction was performed using the Norgen^TM^ (Norgen Biotek, Thorold, ON, Canada) total RNA purification kit as per manufacturer’s instructions. Reverse transcription and preamplification were performed using the TaqMan^®^ (Applied Biosystems, Foster City, CA, USA) OpenArray^®^ MicroRNA Panels protocol. The diluted preamplified product was subject to polymerase chain reaction (PCR) using QuantStudio^TM^ (Applied Biosystems, Foster City, CA, USA). Customised OpenArray^®^ miRNA plates containing the selected miRNA targets were used for the study samples. Data were output from the QuantStudio^TM^ platform as C_RT_ values. The QuantStudio^TM^ platform uses C_RT_, the equivalent of C_t_ [10]. 

### 2.6. Data Analyses

All statistical analyses were performed using the following R packages available from CRAN: BMA (Bayesian Model Averaging) [11]; or Bioconductor version 3.15 (BiocManager 1.30.19): HTqPCR version 1.50.0 [12], in R version R 4.0.0 [13]. For differential expression analysis between time-points, only patients with complete information for all three time-points were analysed. This resulted in 117 individual files with the miRNA C_RT_ value for each patient, corresponding to 39 patients. The HTqPCR package was used to perform Ct data normalization and differential expression analysis as described below. HTqPCR incorporates the empirical Bayes moderated *t*-statistics from the limma package, with the associated *p*-values being used to assess the significance of the observed differential expression [14].

The files were read into R using the HTqPCR function readCtData. Raw data were normalised, and three normalisation methods inside the package were tested: quantile normalisation, deltaCt normalisation, and geometric mean normalisation. Comparing the different normalisation methods, and after the removal of 3 patient samples, the quantile normalisation method performed best at reducing the variation between the data points and was used for the downstream analysis of differential expression. This is illustrated in Appendix A. 

Undetermined values were set to 40 and C_RT_ values above 38 and below 14 were defined as unreliable. Clustering analysis of normalized miRNA data identified two major sample groups that correlated with sample collection dates (Appendix A) according to the QuantStudio^TM^ RT-PCR run batches. Differential expression analysis was performed using the general linear model that takes into account sample collection time-points to correct for possible batch effects. 

Comparisons were made between responders and non-responders for each time-point. The BMA R package was used to identify differentially expressed miRNAs that show a strong correlation to the responder/non-responder status. The Bayesian Model Averaging for generalized linear models function (bic.glm) of the BMA package was run using the 16 miRs of interest and the information about patient response to treatment (good or bad) as a vector, with the following parameters: *glm.family=“binomial”*.

The BiomaRt package [15] in R was used to access the Gene Ontology annotations associated with miRNA targets. The targets were obtained using the multimiR package [16] in R.

Cytoscape was used to produce the interaction network between some of the two miRNA targets found to be significant. AKR1C3 was considered as the gene of interest to start the network. The first neighbours of this gene were retrieved, resulting in only one other gene.

R scripts used in this study are available at https://github.com/GamaPintoLab/Lim-et-al-2023.git.

## 3. Results 

Fifty-two patients were prospectively recruited into this study from 2014–2017. The median patient age was 62 years, with more than two-thirds being male. The majority of patients were diagnosed with stage T3 and node-positive locally advanced rectal cancer. More than two thirds of the patients were of a good performance status. Complete pathological response was seen in 18%. The median follow up was 2.7 years, at which time 21% had recurred. Table 1 summarises the patient characteristics.

### 3.1. CTC Enumeration

A typical image of CTCs captured and the number of CTCs captured and identified using the Isoflux^TM^ instrument and CTC enumeration kit for the 52 patients are shown in Figure 1. The mean CTC count at baseline was 7 CTCs/9 mL blood, ranging from 0 to 45 CTCs/9 mL, with CTCs found to be present in 73% of patients at baseline pre-treatment. The mean CTC count during week 3 of treatment was 5 CTCs/9 mL with a range from 0 to 54/9 mL blood, and the mean count post-treatment or pre-surgery was 6 CTCs/9 mL with a range from 0 to 52/9 mL. The CTC numbers for each individual patient are listed in Appendix A.

### 3.2. miRNA Candidate Selection and miRNA Expression Profiling

A three-tiered selection approach was applied to identify 106 candidate miRNAs of interest. The miRNAs, along with the approach used to select the miRNAs and accompanying references, as well as a data summary, are listed in Appendix A. The distribution of raw miRNA Ct values detected in all samples is shown in the heatmap in Figure 2, revealing the overall detection sensitivity across samples, and the presence of a batch effect that is not corrected by normalization (see Appendix A). Therefore, the next steps of analysis took this into consideration (refer to methods).

### 3.3. Change in CTC and Lymphocyte miRNA Expression during Neoadjuvantchemoradiation

The panel of selected miRNAs used in this study were assessed for differential expression during neoadjuvant treatment as described in the methods. The miRNAs that significantly changed over the course of treatment are listed in Table 2 and shown in Figure 3.

The differential expression analysis between the time-points (baseline vs. during treatment, during treatment vs. post-treatment, and baseline vs. post-treatment) revealed six miRNAs to be differentially expressed between during treatment and post-treatment time-points. In addition, 12 miRNAs were differentially expressed between baseline and during treatment. Of these, three miRNAs (hsa-miR-95, hsa-miR10a, and hsa-miR16-1*) overlapped between the two group comparisons and are shown italicised and in bold in Table 2. No differentially expressed miRNAs were found between the baseline and post-treatment conditions. Thus, between all treatment groups, 15 miRNAs were found to be uniquely differentially expressed. The boxplots for hsa-miR-95, hsa-miR10a, and hsa-miR16-1* miRNAs are represented in Figure 3. For these three differentially expressed overlapping miRNAs there was an apparent increase and then fall in the expression levels back to near baseline levels across the time-points.

### 3.4. Correlation of miRNA Expression with Tumour Response

We next performed a differential expression analysis comparing good responders (TRG 0 and 1) and poor responders (TRG 2 and 3) for each of the time-points. Only one miRNA, hsa-miR-483-5p, was found to be differentially expressed in the post-treatment time-point between good and poor responders, with a fold change of 8.48 (*p* = 0.048), illustrated in Figure 4. miR-19b-3p is also represented in the figure as it was found to be a predictor of good response to treatment (as shown subsequently in Figure 5).

### 3.5. Correlation of miRNA Changes with Tumour Response

To determine if any of the differentially expressed miRNAs are good predictors of treatment response, the dataset was divided into good and poor responders and generalized linear models were generated and ranked by Bayesian modelling average analysis using the BMA package in R [11]. This analysis included the fifteen miRNAs that were found as differentially expressed between the three conditions (pre-treatment, during treatment, and post-treatment) and the single miRNA that was differentially expressed between good responders and poor responders. The independent variables of all the models that passed the BMA cut-off are represented in Figure 5, with Table 3 showing the parameter estimates for the best five models according to the BIC computed and the posterior probabilities associated with each variable.

This analysis showed that there are two miRNAs that belong to the set of differentially expressed miRNAs and show a good correlation to treatment response: hsa-miR-19b-3p and hsa-miR-483-5p, the latter also being the only miRNA found as differentially expressed between good responders and poor responders and showing up as an independent variable in all Bayesian models. This demonstrates a strong correlation of these miRNA with treatment response. Even though miR-19b does not show a significant differential expression between responders and non-responders like miR-483-5p, it is identified as being a good predictor of treatment response. Table 3 summarizes the posterior probabilities for these miRNAs, with hsa-miR-19b-3p and hsa-miR-483-5p shown in bold. Box plot distribution of miR-483-5p and miR-19b-3p expression levels in good responders and poor responders are shown in Figure 4.

### 3.6. Network Analysis and Gene Ontology Annotation

To further investigate the putative impact of dysregulated levels of these miRNAs and the possible functions of their targets and impact in the treatment response, the targets for hsa-miR-19b-3p and hsa-miR-483-5p were retrieved. The gene ontology annotations for these targets were also accessed in R using *biomaRt*. The gene ontology annotations were interrogated for connections with chemoresistance or chemotherapy response. One gene, AKR1C3, a target of miR-483-5p, was found to be annotated as being involved in the metabolism of two drugs commonly used for the treatment of cancer: doxorubicin and daunorubicin. This gene has been previously identified as being overexpressed in several cancers and associated with tumour resistance to chemotherapeutic drugs [17]. The interaction network in Figure 6a shows the connection between some of the hsa-miR-19b-3p and hsa-miR-483-5p target genes having AKR1C3 as a central node. Figure 6B shows the main gene ontology annotations associated with these genes.

## 4. Discussion

This study addresses novel questions which, to our knowledge, have not thus far been reported in the literature. Primarily, this includes the tracking of CTCs at multiple time-points during neoadjuvant treatment for LARC, interrogating CTC samples isolated immunomagnetically for changes in miRNA expression, and, importantly, correlating these with tumour responses.

Presence of CTCs was found in 73% of the patient cohort, which consists of non-metastatic patients. This is a high level for early-stage cancer and is consistent with the Isoflux^TM^ having a greater sensitivity and detection than CellSearch^TM^ [18]. Publications based on CellSearch^TM^ have reported fewer CTCs in the early setting, with the predominance of studies performed in the metastatic setting. Studies utilising other isolation platform, such as fluid-assisted separation technique (FAST), have reported up to 84% having CTCs, the majority consisting of a non-metastatic CRC cohort of which a subgroup consisted of rectal cancers [19]. Another study using a size-based microfluidic device reported CTCs in 100% of rectal cancer patients [20].

CTCs have been found to be higher in rectal cancers treated with neoadjuvant chemoradiation compared to patients who have not received this therapy and there have been variable findings of CTC change with chemoradiation [20,21,22,23,24,25]. Rectal cancers treated with neoadjuvant chemoradiotherapy comprised a small subgroup of one study which enumerated CTCs based on EpCAM positivity [21]. No changes in CTC numbers were found with chemoradiotherapy. On the contrary, another study using a size-based isolation technique found that CTCs fell from a median of 42 CTCs/5 mL pre-chemoradiation to 7 CTCs/5 mL post-treatment [20]. Another study using CellSearch^TM^ found 19% of patients had ≥1 CTC at baseline per 22.5 mL blood [24]. This reduced to 8% post-chemoradiation, 9% within a week of surgery, and 5% at 6 months post-surgery.

We found an overall decrease in CTC numbers over the treatment course, though there is clear heterogeneity between patients. Overall, CTC counts appear to decrease between baseline and during treatment, as well as between baseline and post-treatment with mean CTC numbers post-treatment being slightly higher than during treatment, but lower than at baseline. This is largely consistent with the literature which has reported a fall in CTCs during treatment. Differences in isolation techniques likely account for differences in positivity rate. Importantly, we also looked at three time-points, and tracked CTC change over the treatment course. Most studies have focused on one or two time-points. The increase in CTCs we observed with tumour treatment is similar to another study that also looked at three time-points, albeit all post-treatment. The authors also observed an increase in CTCs within a week of surgery which fell after 6 months. It can be hypothesized that tumour disturbance with treatment or surgery may result in CTC dissemination which results in a transient rise [26].

To our knowledge, there is no literature looking at the miRNA expression profiles of these CTCs and associated lymphocytes. We found that miRNA expression significantly changes during chemoradiation, with 12 differentially expressed miRNAs between baseline and during treatment. Three of these were also differentially expressed from during treatment to post-treatment. Importantly, of these miRNAs, there was a strong correlation between hsa-miR-19b with tumour response, suggesting that low levels of this miRNA correlate with a good response to chemoradiation.

hsa-miR-95, hsa-miR-10a, and hsa-miR-16-1 have all been implicated in colorectal cancer carcinogenesis. These three miRNAs were found in our study to be significantly differentially expressed across treatment time-points. miR-95 is thought to target sorting nexin 1 (SNX1) and promotes cell proliferation and oncogenesis [27]. miR-95-3p has also been shown to regulate cell survival through the downregulation of HDGF [28]. miR-10 has been shown to be upregulated in rectal cancer tissue [29], while miR-16 has been associated with prediction of tumour responses [30].

This finding of miR-19b and correlation with tumour response is supported by other studies in the literature, mainly in solid tissue [31,32]. miR-19b expression levels in LARC biopsies obtained at time of colonoscopy have been identified as an independent predictor of pathological response to preoperative chemoradiation. miR-19b expression increases in chemotherapy-resistant cells and downregulation of miR-19b enhances the antitumor effects of chemotherapy [32]. miR-19b was also found to be overexpressed in colorectal tissue, and indicated a poor prognosis. Another study found that high expression of miR-19b in colorectal cancer was associated with lymph node metastasis and distant metastasis [33]. FBXW7 has been identified as a putative target of miR-19b, and miR-19b inhibition has been shown to potentially enhance the efficacy of radiotherapy [34]. Another potential target of miR-19b is ITGB8, with findings suggesting that the miR19b-3p/ITGB8 axis plays an important role in the growth and metastasis of colorectal [35]. Taken together, low levels of miR-19b expression in CTCs are associated with good response to chemoradiation and favourable outcomes. miR-19b thus represents a potential predictive biomarker as well as a potential strategy to sensitise tumour cells to treatment.

Another promising miRNA candidate is miRNA hsa-miR-483-5p, which was differentially expressed between good responders and poor responders and showed up as an independent variable in all our Bayesian models. This is consistent with the literature of this miRNA in solid tissue, where upregulated miR-483-5p was found in TRG1 patients (good responder) compared to TRG > 1 (poor responder) patients as classified by the Mandard TRG classification [36]. Similar findings are shown in another study, where this miRNA candidate correlated with chemoradiation response [37]. It was also reported that miR-483 expression may also be linked to *IGF2* and carcinogenesis, possibly making it a potential diagnostic biomarker [38].

The gene ontology annotations for these two miRNAs are shown above and they may play a role in the Wnt and MAPK pathways. miR-483-5p is believed to regulate AKR1C3. The upregulation of hsa-miR-483-5p in responders when compared to non-responders could indicate that the targeting of AKR1C3 is contributing to a better response to the drug, which makes this miRNA a potential clinically useful predictor of response to treatment.

In summary, the additional information from liquid biopsies measured during treatment of LARC is invaluable, as response prediction during chemoradiation may serve to either intensify or de-escalate treatment. This is particularly relevant due to the move toward total neoadjuvant therapy and the watch and wait approach, potential alternatives to shift the traditional paradigm of chemoradiation and surgery [2,3]. The next steps would be to validate these findings in a larger study, and to further investigate the functions of these miRNAs in cell lines using RNA overexpression or RNA mediated interference to silence the genes of interest.

## 5. Conclusions

Our data show that CTCs were present in the vast majority of non-metastatic LARC patients, and fell during chemoradiation, although there was heterogeneity between patients. To our knowledge, this is the first study to report miRNA expression profiles from isolated CTCs in LARC rather than from solid tissue, and we found that expression changes during treatment, with three miRNAs being differentially expressed across the three time-points, and a further twelve miRNAs were differentially expressed in pairwise time-point comparisons. miR-19b-3p and miR-483-5p correlated with good response, the latter also found to be differentially expressed between good responders and poor responders. These findings are largely supportive of the literature and the putative targets of these miRNAs. Thus, they represent potential predictive biomarkers as well as a potential strategy to sensitise tumour cells to treatment.

## Figures and Tables

**Figure 1 cancers-15-04184-f001:**
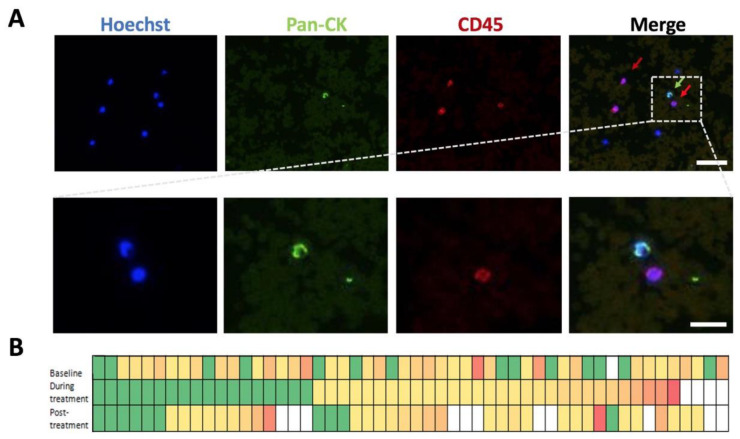
(**A**). Enumeration of circulating tumour cells (CTCs) from a patient isolated using the Isoflux^TM^ instrument. The upper panel shows a CTC (green arrow) and lymphocytes (red arrow) and the lower panel is an enlargement of the area indicated. The identified CTC is DAPI-positive, pan-CK positive, and CD45 negative. Immunomagnetic beads can be seen in the background. Scale bar: 50 µm upper panel, 20 µm lower panel. (**B**). Heatmap representing the CTCs identified from 52 patients at baseline, during treatment and post-treatment. These time-points have 1, 4, and 11 samples missing respectively (represented by the white boxes). Each column on the X-axis represents a patient. Green represents low numbers with orange to red colour gradient representing increasing numbers of CTCs.

**Figure 2 cancers-15-04184-f002:**
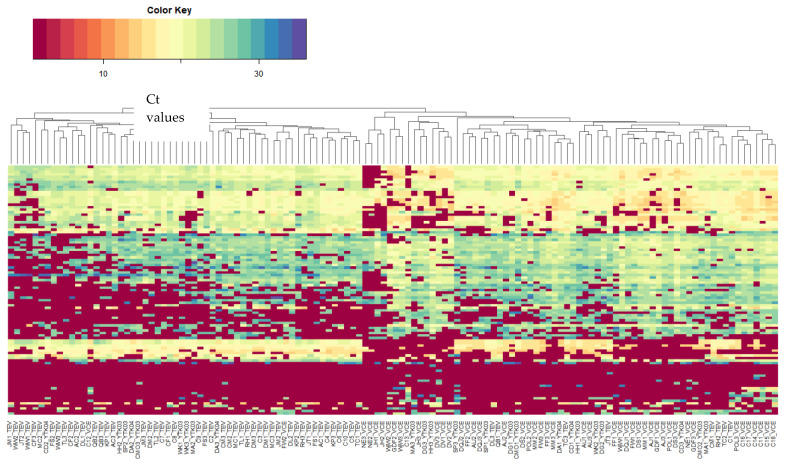
Heatmap of raw microRNA Ct values detected in all samples from the 52 patients (columns on X-axis). The first two letters in the sample name identify the patient; the number that follows the patient code identifies the condition (1—baseline; 2—during treatment; 3—post-treatment). The following codes (TEV, VCE, and YTK) identify the batch of the samples, i.e., the time of sample collection and analysis.

**Figure 3 cancers-15-04184-f003:**
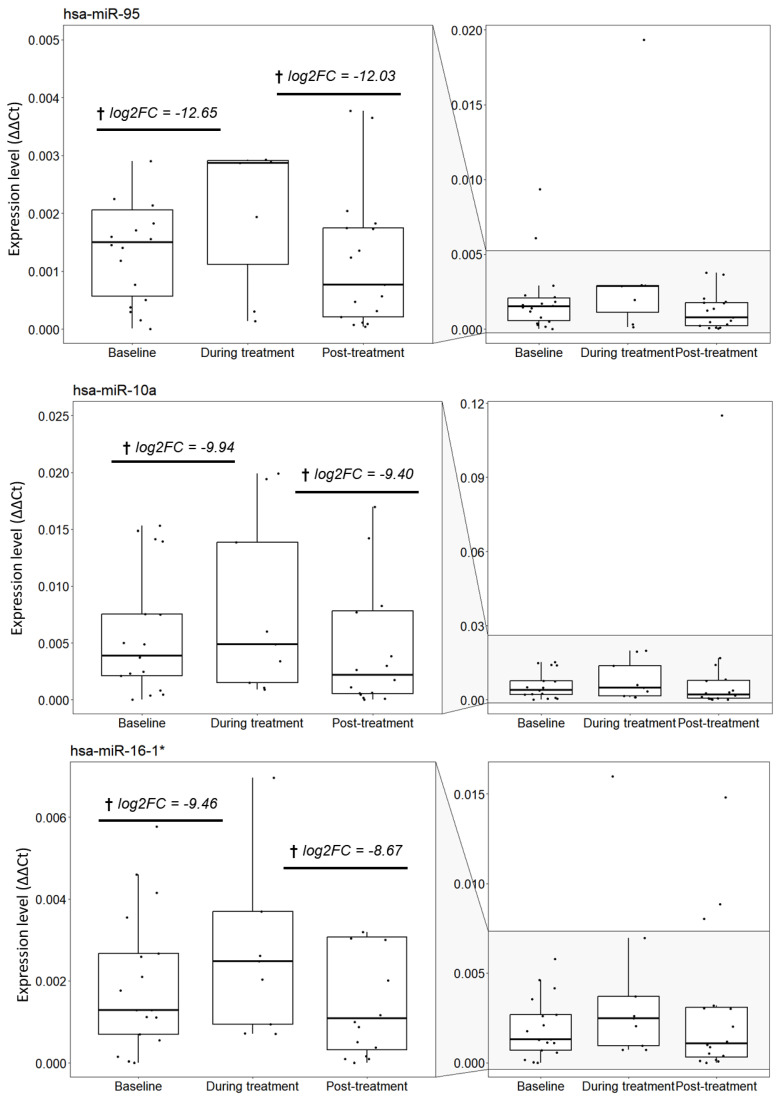
Box plots showing the three microRNAs that are significantly differentially expressed between during treatment vs. post-treatment, as well as between baseline vs. during treatment. The expression level was calculated using the quantile normalisation values of microRNAs and reference gene to calculate relative expression ∆∆Ct for each patient in each condition. † log2FC values corresponding to during treatment vs. baseline and during treatment vs. post-treatment comparisons, respectively.

**Figure 4 cancers-15-04184-f004:**
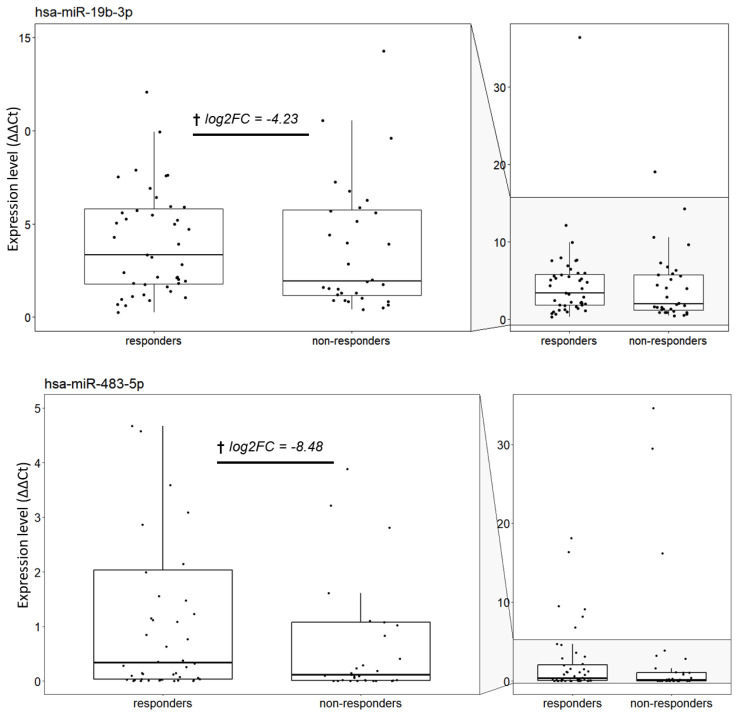
Box plot distribution of miR-483-5p and miR-19b-3p expression levels in good responders and poor responders. The expression level was calculated using the quantile normalisation values of microRNAs and reference genes to calculate relative expression ∆∆Ct for each patient in each condition. † log2FC values corresponding to responders vs. non-responders to treatment comparisons.

**Figure 5 cancers-15-04184-f005:**
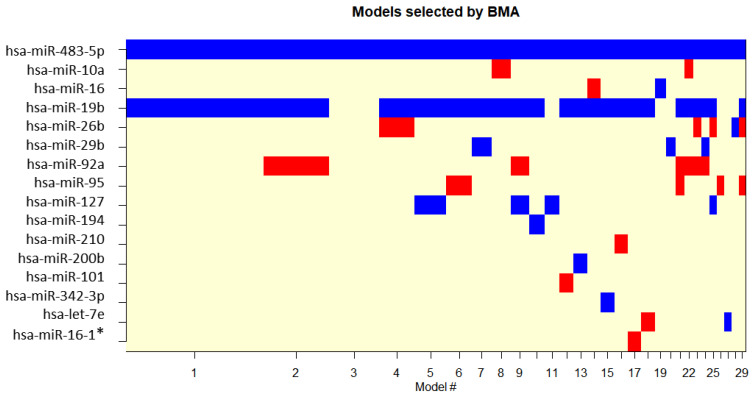
Cumulative posterior model probabilities resulting from the BMA package. Red colour displays negative variable estimate, blue colour displays positive variable estimate, uncoloured variables were not included in the models. The models are ordered according to the decreasing posterior model probability.

**Figure 6 cancers-15-04184-f006:**
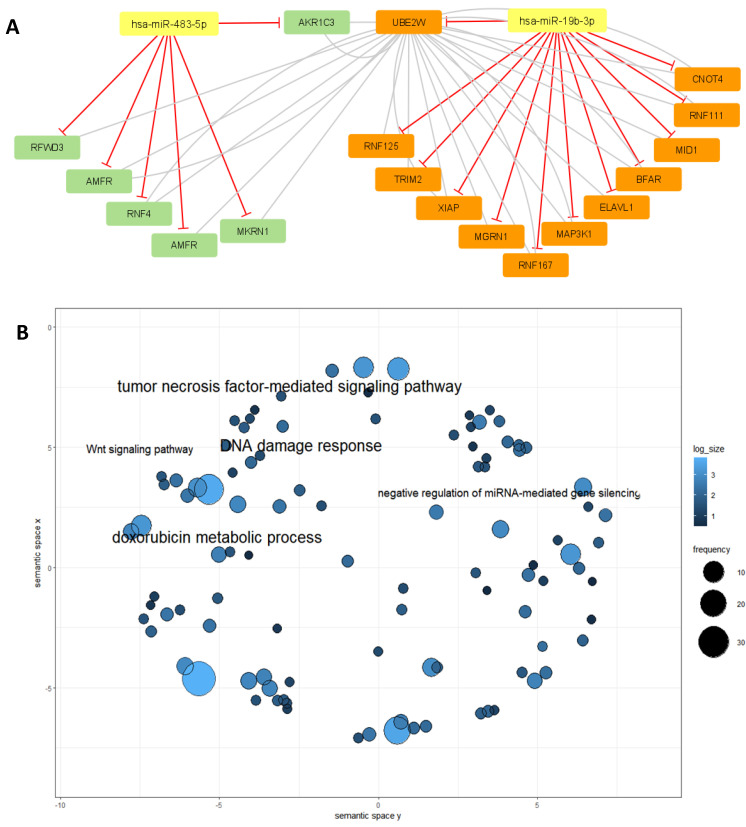
(**A**) Network representation of some of the hsa-miR-483-5p and hsa-miR-19b-3p targets. To obtain this reduced network, miR-483-5p-target AKR1C3’s first neighbours were selected resulting in UBE2W. Afterwards, the first neighbours of UBE2W were selected to create the present network. The targets of miR-483-5p are represented in green, and the targets for miR-19b in orange. (**B**) Gene ontology annotation using the genes present in the network. Besides the connection to the metabolism of anthracycline doxorubicin, other important functions such as the Wnt signalling pathway or the negative regulation by microRNAs have been identified using these genes.

**Table 1 cancers-15-04184-t001:** Clinicopathological characteristics of the patient cohort.

Patient and Tumour Characteristics	n	Percent (%)
**Age**		
˂65	30	58
≥65	22	42
**Gender**		
Male	38	73
Female	14	27
**Ethnicity**		
Caucasian	37	71
Asian	15	29
**Tumour stage ***		
T2	5	10
T3	42	80
T4	5	10
**Nodal stage ***		
N0	8	15
N1	20	38
N2	24	47
**Smoking status**		
Yes	22	42
No	30	58
**ECOG performance status**		
0	37	71
1	15	29
**Tumour regression grade ^**		
0	9	18
1	15	31
2	21	43
3	4	8
**Recurrence**		
Yes	11	21
No	41	79

Abbreviations: T = tumour, N = nodal, * based on AJCC classification, ^ 3 patients did not undergo surgery and hence do not have a TRG grade.

**Table 2 cancers-15-04184-t002:** Differentially expressed microRNAs from circulating tumour cells and associated lymphocytes at baseline, during treatment, and post-treatment.

miRNA	Log2 Fold Change	Adjusted *p*-Value
**Baseline and during treatment**		
** *hsa-miR-95* **	** *12.65* **	** *0.00031* **
** *hsa-miR-10a* **	** *9.94* **	** *0.0045* **
** *hsa-miR-16-1** **	** *9.46* **	** *0.0087* **
hsa-miR-210	9.31	0.0045
hsa-miR-194	9.086	0.016
hsa-miR-101	8.76	0.038
hsa-miR-29b	8.56	0.0046
hsa-miR-19b	−4.60	0.016
hsa-miR-342−3p	−4.55	0.021
hsa-miR-16	−4.19	0.014
hsa-miR-92a	−3.25	0.034
hsa-miR-26b	−2.88	0.038
**During treatment and post-treatment**		
hsa-let-7e	4.32	0.020
** *hsa-miR-95* **	** *−12.03* **	** *0.00086* **
** *hsa-miR-10a* **	** *−9.40* **	** *0.010* **
** *hsa-miR-16-1** **	** *−8.67* **	** *0.020* **
hsa-miR-200b	−7.95	0.010
hsa-miR-127	−4.45	0.020

The top section shows the twelve microRNAs that are differentially expressed between baseline and during treatment, while the bottom section shows the six microRNAs that are differentially expressed between during treatment and post-treatment. The log transformed-fold changes are listed with corresponding adjusted *p*-values. The microRNAs in common across the two comparisons are italicised and in bold.

**Table 3 cancers-15-04184-t003:** Summary output for the best five models and corresponding predictors of treatment response.

	p! = 0	EV	SD	Model 1	Model 2	Model 3	Model 4	Model 5
Intercept	100	6.97	5.094072	9.30181	3.34419	2.89069	4.60466	12.00436
*** hsa-miR-483-5p**	100	−1.21 × 10^−1^	0.035075	−0.12257	−0.12391	−0.11371	−0.12577	−0.12061
hsa-miR-10a	4.5	1.65 × 10^−3^	0.012055					
hsa-miR-16	4	−1.96 × 10^−3^	0.047242					
*** hsa-miR-19b**	82.7	−3.65 × 10^−1^	0.286371	−0.36668	−0.6098		−0.57137	−0.32431
hsa-miR-26b	10.5	3.48 × 10^−2^	0.147071				0.40815	
hsa-miR-29b	5.9	−2.25 × 10^−3^	0.013898					
hsa-miR-92a	19.1	9.45 × 10^−2^	0.232413		0.50124			
hsa-miR-95	7.9	4.54 × 10^−3^	0.021901					
hsa-miR-127	11.7	−1.12 × 10^−2^	0.039798					−0.09133
hsa-miR-194	2.5	−4.01 × 10^−4^	0.005315					
hsa-miR-210	2.2	7.53 × 10^−5^	0.004704					
hsa-miR-200b	2.2	−3.92 × 10^−4^	0.013972					
hsa-miR-101	2.3	4.21 × 10^−4^	0.009652					
hsa-miR-342-3p	2.2	−2.66 × 10^−4^	0.012814					
hsa-let-7e	3.4	−4.97 × 10^−4^	0.014303					
hsa-miR-16-1	2.2	9.23 × 10^−5^	0.00867					
nVar				2	3	1	3	3
BIC				−353.513	−352.046	−351.483	−350.8	−350.573
post prob				0.222	0.107	0.08	0.057	0.051

p! = 0—posterior probability that the variable is in the model (%); EV—BMA posterior mean; SD—BMA posterior standard deviation for each variable. The two microRNAs that have the best performance in the model showing up as possible good predictors of response to treatment are bolded and asterisked.

## Data Availability

R scripts used in this study are available at https://github.com/GamaPintoLab/Lim-et-al-2023.git.

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
