# Peer review of "Circulating Tumour Cell Associated MicroRNA Profiles Change during Chemoradiation and Are Predictive of Response in Locally Advanced Rectal Cancer"

_cancers, 2023, doi:10.3390/cancers15164184_

Round 1

Reviewer 1 Report

The paper is well written and constructed. It is of interest.

I am not a good judge for array analysis and statistical analysis.

Figure 4 shows patients with very high levels of has-miR-483-5p in the non-responders group. Their situation should be discussed.

The authors could discuss if it is possible to measure miRNA levels directly in the blood of patients instead of extracting them from purified CTCs.

Specific points in manuscript:

-          The quality of the figures is poor. Writings are blurry (figure 2, 2, 5, and 6)

-          The scale bar of the second line of images in figure 1A.

-          The lymphocyte of CD45 image in the first line in figure 1A is not visible.

-          In the legend of figure 1A one should include in brakets the values considered for “low, medium and high” number of CTCs.

-          The color code in figure 6B should be indicated.

Reviewer 2 Report

Authors Lim, Chua, Ng, Ip, Marques, Tran, Gama-Carvalho, Asghari, Henderson, Ma, Souza and Spring report on potential miRNAs associated with circulating tumor cells to be used as ‘biological-markers’ to be used as a proxy to assess treatment and remission.

Unfortunately, this reviewer suggests ‘to reject the paper’, due to the unconvincing data. Below are points for reason of rejection:  

-line 216-7 states “No differentially expressed miRNAs were found between the baseline and post-treatment conditions”. This data would suggest that treatment does not change miRNA expression but rather just slightly alters abundance of circulating miRNAs. Thus, it is unclear the relevance of the remaining data.

-authors state ‘statistical significance’ for data within Table 2 and Figure 3 but the statistical test used to make this claim cannot be found within the paper. The only part possibly associated with statistical analysis is that the authors state they used an R package. But given the wide range of variables one can query and change with any package within R, how is the reader to feel confident the analysis was done correctly.

-authors use BMA R package but do not include a script or even parameters for how analysis was performed, which suggests to this reviewer the limitations in the authors' understanding of how to use this statistical program in R.

-Figure 1A,

the images are not clear

-Figure 1B,

what do the white boxes represent?

Also, if the goal is to show that treatment affects CTC number, why not graph the data (with an R package or excel or graphpad…) to visually illustrate the potential temporal changes of CTCs before, during, and post-treatment. The ‘heatmap’, where the authors did not even state how the gradient was determined, does not illustrate clearly the ‘change’ of CTCs.

-Figure 2,

-due to the hierarchical clustering of miRNA expression associated with patients, the graph does not clearly illustrate any potential change in miRNA expression correlating with baseline, during, post-treatment the hierarchical clustering. This reviewer suggests the authors should group the patients (on the y-axis) based on temporal treatment (baseline, during, and post), and then assess possible miRNA expression trends. As the data are illustrated now with hierarchical clustering of miRNA expression, the graph is misleading.

-The y-axis is un-readable.

-RT-qPCR data should never be illustrated by ‘Ct values’. This type of expression data has to be normalized to an internal control gene whose expression does not change.

-Figures 4 and 5

the miRNA expression associated with ‘responders’ is the only data that looks somewhat convincing but the authors do not show how the R-packages were run, a script should be supplied in supplement for reproducibility purposes.

-Figure 6,

miRNA prediction programs alone are not convincing to illustrate biological significance. Most mRNA target predictions generate a high number of false-positives. Thus, most researchers follow this type of analysis with verification in cell lines. Regarding 6B, what biology is not associated with MAPK and Wnt signaling? This whole figure lacks substance.  

Lastly, the data are based on RT-qPCR data of known miRNAs. Why not perform small-RNA-Seq to assess miRNA abundance in a non-biased manner over probe based methodology? Also, small-RNA-Seq libraries would illuminate changes in various classes of small-RNAs, thus elucidating any potential ‘biomarkers’ associated with treatment.

english seems fine

Reviewer 3 Report

In this manuscript the Authors investigate possible biomerkers of the outcome of trimodality treatment for locally advanced rectal cancer (LARC): circulating tumor cells and microRNAs. Changes in expression of microRNAs after a  neoadjuvant chemoradiation for LARC were analysed the study.

In a prospective determination in 52 LARC-patients, a correlation of microRNAs hsa-miR-19b-3p and hsa-miR-483-5p in circulating tumor cells with a good response to treatment was observed in a follow up. The Authors propose that microRNA profiles may be potential predictors of response to treatment.

This intelligent novel study is well designed and properly conducted using up-to-date mollecular determinations.

The results of this paper may add new knowledge about mechanisms of a poor outcome observed in a third LARC-patients despite today's treatment. I recommend publishing this paper.

Reviewer 4 Report

The authors' hypothesis that circulating tumor cells and associated microRNA profiles change with chemoradiation therapy, and that these changes predict overall response in locally advanced colorectal cancer, is intriguing. However, the data presentation makes their findings difficult to interpret.

I recommend including a figure that clearly shows fold change in hsa-miR-95, hsa-miR-10a, and hsa-miR-16-1 expression at baseline, during treatment, and post-treatment (the current figure is hard to interpret). Additionally, a graph demonstrating the differential fold change between responders and non-responders for hsa-miR-483-5p and hsa-miR-19b-3p would better highlight these results.

The discussion section would also benefit from more information about the roles of hsa-miR-95, hsa-miR-10a, and hsa-miR-16-1 and how the observed expression changes may relate to response to chemoradiation therapy in colorectal cancer. Improved data visualization and expanded discussion of the key miRNAs would strengthen the study conclusions.

Round 2

Reviewer 2 Report

Decision is still 'reject'. 

The reviewer appreciates the addition of code to a Github account which should be standard now for most labs that use any software developed by 3rd parties. 

However, most of the comments were not adequately addressed, some examples below. 

-raw Ct values are not appropriate

-authors ignored comments on Figure 6

-authors ignored comments on Figure 2 on grouping patients based on treatment to look for trends. The clustering shows trends but what the trends represent is unclear. 

na

Reviewer 4 Report

The authors have address the comments

Author Response

Thank you for acknowledging our changes as satisfactory.